# INCREMENTAL LEARNING THROUGH DEEP ADAPTATION

## ABSTRACT

Given an existing trained neural network, it is often desirable to learn new capabilities without hindering performance of those already learned. Existing approaches either learn sub-optimal solutions, require joint training, or incur a substantial increment in the number of parameters for each added task, typically as many as the original network. We propose a method called *Deep Adaptation Networks* (DAN) that constrains newly learned filters to be linear combinations of existing ones. DANs preserve performance on the original task, require a fraction (typically 13%) of the number of parameters compared to standard fine-tuning procedures and converge in less cycles of training to a comparable or better level of performance. When coupled with standard network quantization techniques, we further reduce the parameter cost to around 3% of the original with negligible or no loss in accuracy. The learned architecture can be controlled to switch between various learned representations, enabling a single network to solve a task from multiple different domains. We conduct extensive experiments showing the effectiveness of our method on a range of image classification tasks and explore different aspects of its behavior.

## 1 INTRODUCTION

While deep neural networks continue to show remarkable performance gains in various areas such as image classification (Krizhevsky et al. (2012)), semantic segmentation (Long et al. (2015)), object detection (Girshick et al. (2014)), speech recognition (Hannun et al. (2014))medical image analysis (Litjens et al. (2017)) - and many more - it is still the case that typically, a separate model needs to be trained for each new task. Given two tasks of a totally different modality or nature, such as predicting the next word in a sequence of words versus predicting the class of an object in an image, it stands to reason that each would require a different architecture or computation. However, for a set of related tasks such as classifying images from different domains it is natural to expect that solutions will (1) Utilize the same computational pipeline; (2) Require a modest increment in the number of required parameters for each added task; (3) Be learned without hindering performance of already learned tasks (a.k.a catastrophic forgetting) and (4) Be learned incrementally, dropping the requirement for joint training such as in cases where the training data for previously learned tasks is no longer available.

Our goal is to enable a network to learn a set of related tasks one by one while adhering to the above requirements. We do so by augmenting a network learned for one task with *controller modules* which utilize already learned representations for another. The parameters of the controller modules are optimized to minimize a loss on a new task. The training data for the original task is not required at this stage. The network's output on the original task data stays exactly as it was; any number of controller modules may be added to each layer so that a single network can simultaneously encode multiple distinct tasks, where the transition from one task to another can be done by setting a binary switching variable or controlled automatically. The resultant architecture is coined DAN, standing for **D**eep **A**daptation **N**etworks. We demonstrate the effectiveness of our method on the recently introduced Visual Decathlon Challenge (Rebuffi et al. (2017)) whose task is to produce a classifier to work well on ten different image classification datasets. Though adding only 13% of the number of original parameters for each newly learned task (the specific number depends on the network architecture), the average performance surpasses that of fine tuning *all* parameters - without the negative side effects of doubling the number of parameters and catastrophic forgetting. In this work,

we focus on the task of image classification on various datasets, hence in our experiments the word "task" refers to a specific dataset.

Our main contribution is the introduction of an improved alternative to transfer learning, which is as effective as fine-tuning all network parameters towards a new task, precisely preserves old task performance, requires a fraction (network dependent, typically 13%) of the cost in terms of new weights and is able to switch between any number of learned tasks.

We introduce two variants of the method, a fully-parametrized version, whose merits are described above and one with far fewer parameters, which significantly outperforms shallow transfer learning (i.e. feature extraction) for a comparable number of parameters. In the next section, we review some related work. Sec. 3 details the proposed method. In Sec. 4 we present various experiments, including comparison to related methods, as well as exploring various strategies on how to make our method more effective, followed by some discussion & concluding remarks.

## 2 RELATED WORK

**Multi-task Learning.** In multi-task learning, the goal is to train one network to perform several tasks simultaneously. This is usually done by jointly training on all tasks. Such training is advantageous in that a single representation is used for all tasks. In addition, multiple losses are said to act as an additional regularizer. Some examples include facial landmark localization (Zhu et al. (2015)), semantic segmentation (He et al. (2017)), 3D-reasoning (Eigen and Fergus (2015)), object and part detection (Bilen and Vedaldi (2016)) and others. While all of these learn to perform different tasks on the same dataset, the recent work of (Bilen and Vedaldi (2017)) explores the ability of a single network to perform tasks on various image classification datasets. We also aim to classify images from multiple datasets but we propose doing so in a manner which learns them one-by-one rather than jointly. Concurrent with our method is that of Rebuffi et al. (2017) which introduces dataset-specific additional residual units. We compare to this work in Sec 4. Our work bears some resemblance to Misra et al. (2016), where two networks are trained jointly, with additional "cross-stitch" units, allowing each layer from one network to have as additional input linear combinations of outputs from a lower layer in another. However, our method does not require joint training and requires significantly fewer parameters.

**Incremental Learning.** Adding a new ability to a neural net often results in so-called "catastrophic forgetting" (French (1999)), hindering the network's ability to perform well on old tasks. The simplest way to overcome this is by fixing all parameters of the network and using its penultimate layer as a feature extractor, upon which a classifier may be trained (Donahue et al. (2014); Sharif Razavian et al. (2014)). While guaranteed to leave the old performance unaltered, it is observed to yield results which are substantially inferior to fine-tuning the entire architecture (Girshick et al. (2014)). The work of Li and Hoiem (2016) provides a succinct taxonomy of several variants of such methods. In addition, they propose a mechanism of fine-tuning the entire network while making sure to preserve old-task performance by incorporating a loss function which encourages the output of the old features to remain constant on newly introduced data. While their method adds a very small number of parameters for each new task, it does not guarantee that the model retains its full ability on the old task. In Rusu et al. (2016) new representations can be added alongside old ones while leaving the old task performance unaffected. However, this comes at a cost of duplicating the number of parameters of the original network for each added task. In Kirkpatrick et al. (2017) the learning rate of neurons is lowered if they are found to be important to the old task. Our method *fully* preserves the old representation while causing a modest increase in the number of parameters for each added task.

**Network Compression** Multiple works have been published on reducing the number weights of a neural network as a means to represent it compactly (Han et al. (2015a;b)), gain speedups (Denton et al. (2014)) or avoid over-fitting (Hanson and Pratt (1988)), using combinations of coding, quantization, pruning and tensor decomposition. Such methods can be used in conjunction with ours to further improve results, as we show in Sec. 4.

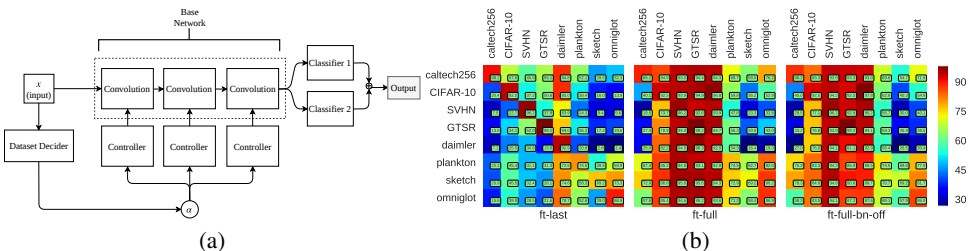

(a)             (b)

Figure 1: (a) Summary of proposed method. Each convolutional layer of a base network is modified by re-combining its weights through a controller module. A set of controllers is added for each newly learned task. A binary switching vector $\alpha$ controls the output of the network by switching the controller modules on or off. $\alpha$ can be determined either manually or via a sub-network ("Dataset Decider") which determines the source domain of the image, switching accordingly between different sets of control parameters. Other layers (e.g, non-linearities, batch-normalization, skip layers) not shown for presentation purposes. (b) Transferability of various datasets to each other (*ft-last*) fine tuning only the last layer (*full*) fine-tuning all layers (*ft-full-bn-off*) fine tuning all layers while disallowing batch-normalization layers' weights to be updated. Overall, networks tend to be more easily transferable to problems from related domain (e.g., natural / drawing). Zoom in to see numbers. It is recommended to view this figure in color on-line.

## 3    APPROACH

We begin with some notation. Let $T$ be some task to be learned. Specifically, we use a deep convolutional neural net (DCNN) in order to learn a classifier to solve $T$, which is an image classification task. Most contemporary DCNN's follow a common structure: for each input $x$, the DCNN computes a representation of the input by passing it through a set of $l$ layers $\phi_i$, $i \in 1 \ldots l$ interleaved with non-linearities. The initial (lower) layers of the network are computational blocks, e.g. convolutions with optional residual units in more recent architectures (He et al. (2016)). Our method applies equally to networks with or without residual connections. At least one fully connected layer $f_i$, $i \in 1 \ldots c$ is attached to the output of the last convolutional layer. Let $\Phi_{F_N} = \sigma(\phi_l) \circ \ldots \sigma(\phi_2) \circ \sigma(\phi_1)$ be the composition of all of the convolutional layers of the network $N$, interleaved by non-linearities. We use an architecture where all non-linearities $\sigma$ are the same function, with no tunable parameters. Denote by $\Phi_{F_N}(x)$ the *feature* part of $N$. Similarly, denote by $\Phi_{C_N} = f_c \circ \ldots \sigma(f_2) \circ \sigma(f_1)$ the *classifier* part of $N$, i.e. the composition of all of the fully-connected layers of $N$. The output of $N$ is then simply defined as:

$$N(x) = \Phi_{C_N} \circ \Phi_{F_N}(x) \tag{1}$$

We do not specify batch-normalization layers in the above notation for brevity. It is also possible to drop the $\Phi_{C_N}$ term, if the network is fully convolutional, as in Long et al. (2015).

### 3.1    ADAPTING REPRESENTATIONS

Assume that we are given two tasks, $T_1$ and $T_2$, to be learned, and that we have learned a *base network N* to solve $T_1$. We assume that a good solution to $T_2$ can be obtained by a network with the same architecture as $N$ but with different parameters. We augment $N$ so that it will be able to solve $T_2$ as well by attaching a controller module to each of its convolutional layers. Each controller module uses the existing weights of the corresponding layer of $N$ to create new convolutional filters adapted to the new task $T_2$: for each convolutional layer $\phi_l$ in $N$, let $F_l \in \mathcal{R}^{C_o \times C_i \times k \times k}$ be the set of filters for that layer, where $C_o$ is the number of output features, $C_l$ the number of inputs, and $k \times k$ the kernel size (assuming a square kernel). Denote by $b_l \in \mathcal{R}^C$ the bias. Denote by $\tilde{F}_l \in \mathcal{R}^{C_o \times D}$ the matrix whose rows are flattened versions of the filters of $F_l$, where $D = C_i \cdot k \cdot k$; let $f \in \mathcal{R}^{C_i \times k \times k}$ be a filter from $F_l$ whose values are $f^1 = \begin{pmatrix} f_{11}^1 & \cdots & f_{1k}^1 \\ & \ddots & \\ & & f_{kk}^1 \end{pmatrix}, \cdots, f^i = \begin{pmatrix} f_{11}^i & \cdots & f_{1k}^i \\ & \ddots & \\ & & f_{kk}^i \end{pmatrix}.$

The flattened version of $f$ is a row vector $\tilde{f} = (f_{11}^1, \cdots, f_{kk}^1, \cdots, \cdots f_{11}^i, \cdots, f_{kk}^i) \in \mathcal{R}^{\mathcal{D}}$. "Unflattening" a row vector $\tilde{f}$ reverts it to its tensor form $f \in \mathcal{R}^{C_i \times k \times k}$. This way, we can write

$$\tilde{F}_l^a = W_l \cdot \tilde{F}_l \tag{2}$$

where $W_l \in \mathcal{R}^{C_o \times C_o}$ is a weight matrix defining linear combinations of the flattened filters of $F_l$, resulting in $C_o$ new filters. Unflattening $\tilde{F}_l^a$ to its original shape results in $F_l^a \in \mathcal{R}^{C_o \times C_i \times k \times k}$, which we call the adapted filters of layer $\phi_l$. Using the symbol $X \otimes Y$ as shorthand for *flatten* $Y \rightarrow$ *matrix multiply by* $X \rightarrow$ *unflatten*, we can write:

$$F_l^a = W_l \otimes F_l \tag{3}$$

If the convolution contains a bias, we instantiate a new weight vector $b_l^a$ instead of the original $b_l$. The output of layer $\phi_l$ is computed as follows: let $x_l$ be the input of $\phi_l$ in the adapted network. For a given switching parameter $\alpha \in \{0, 1\}$, we set the output of the modified layer to be the application of the switched convolution parameters and biases:

$$x_{l+1} = [\alpha(W_l \otimes F_l) + (1 - \alpha)F_l] * x_l + \alpha b_l^a + (1 - \alpha)b_l \tag{4}$$

A set of fully connected layers $f_i^a$ are learned from scratch, attaching a new "head" to the network for each new task. Throughout training & testing, the weights of $F$ (the filters of $N$) are kept fixed and serve as basis functions for $F^a$. The weights of the controller modules are learned via backpropagation given the loss function. Weights of any batch normalization (BN) layers are either kept fixed or learned anew. The batch-normalized output is switched between the values of the old and new BN layers, similarly to Eq. 4. A visualization of the resulting DAN can be seen in Fig. 1.

**Weaker parametrization** A weaker variant of our method is one that forces the matrices $W_l$ to be diagonal, e.g, only scaling the output of each filter of the original network. We call this variant "*diagonal*" (referring only to scaling coefficients, such as by a diagonal matrix) and the full variant of our method "*linear*" (referring to a linear combination of filters). The diagonal variant can be seen as a form of explicit regularization which limits the expressive power of the learned representation. While requiring significantly fewer parameters, it results in poorer classification accuracy, but as will be shown later, also outperforms regular feature-extraction for transfer learning, especially in network compression regimes.

**Multiple Controllers** The above description mentions one base network and one controller network. However, any number of controller networks can be attached to a single base network, regardless of already attached ones. In this case $\alpha$ is extended to one-hot vector of values determined by another sub-network, allowing each controller network to be switched on or off as needed.

In the following, we denote a network learned for a dataset/task $S$ as $N_S$. A controller learned using $N_S$ as a base network will be denoted as $DAN_S$, where DAN stands for **D**eep **A**daptation **N**etwork and $DAN_{S \rightarrow T}$ means using $DAN_S$ for a specific task $T$. While in this work we apply the method to classification tasks it is applicable to other tasks as well.

**Parameter Cost** The number of new parameters added for each task depends on the number of filters in each layer and the number of parameters in the fully-connected layers. As the latter are not reused, their parameters are fully duplicated. Let $M = C_o \times D$ be the filter dimensions for some conv. layer $\phi_l$ where $D = C_i \times k \times k$. A controller module for $\phi_l$ requires $C_o^2$ coefficients for $F_l^a$ and an additional $C_o$ for $b_l^a$. Hence the ratio of new parameters w.r.t to the old for $\phi_l$ is $\frac{C_o \times (C_o + 1)}{C_o \times (D + 1)} = \frac{C_o + 1}{D + 1} \approx \frac{C_o}{D}$. Example: for $C_o = C_i = 256$ input and output units and a kernel size $k = 5$ this equals $\frac{256 + 1}{256 \cdot 5^2 + 1} \approx 0.04$. In the final architecture we use the total number of weights required to adapt the convolutional layers $\Phi_l$ combined with a new fully-connected layer amounts to about 13% of the original parameters. For VGG-B, this is roughly 21%. For instance, constructing 10 classifiers using one base network and 9 controller networks requires $(1 + 0.13 * 9) \cdot P$=2.17 $\cdot P$ parameters where $P$ is the number for the base network alone, compared to $10 \cdot P$ required to train each network independently. The cost is dependent on network architecture, for example it is higher

| Net | C-10 | GTSR | SVHN | Caltech | Dped | Oglt | Plnk | Sketch | Perf. | #par |
|---|---|---|---|---|---|---|---|---|---|---|
| **VGG-B(S)** | 92.5 | 98.2 | 96.2 | 88.2 | 92.9 | 86.9 | 74.5 | 69.2 | 87.32 | 8 |
| **VGG-B(P)** | 93.2 | 99.0 | 95.8 | 92.6 | 98.7 | 83.8 | 73.2 | 65.4 | 87.71 | 8 |
| **DAN$_{caltech-256}$** | 77.9 | 93.6 | 91.8 | 88.2 | 93.8 | 81.0 | 63.6 | 49.4 | 79.91 | 2.54 |
| $DAN_{sketch}$ | 77.9 | 93.3 | 93.2 | 86.9 | 94.0 | 85.4 | 69.6 | 69.2 | 83.7 | 2.54 |
| $DAN_{noise}$ | 68.1 | 90.9 | 90.4 | 84.6 | 91.3 | 80.6 | 61.7 | 42.7 | 76.29 | 1.76 |
| $DAN_{imagenet}$ | 91.6 | 97.6 | 94.6 | 92.2 | 98.7 | 81.3 | 72.5 | 63.2 | 86.46 | 2.76 |
| $DAN_{imagenet+sketch}$ | 91.6 | 97.6 | 94.6 | 92.2 | 98.7 | 85.4 | 72.5 | 69.2 | 87.76 | 3.32 |

Table 1: Perf: top-1 accuracy (%, higher is better) on various datasets and parameter cost (#par., lower is better) for a few baselines and several variants of our method. Rows 1,2: independent baseline performance. *VGG-B*: VGG (Simonyan and Zisserman (2014)) architecture B. (S) - trained from scratch. (P) - pre-trained on ImageNet. Rows 3-7: (ours) controller network performance; $DAN_{sketch}$ as a base network outperforms $DAN_{caltech-256}$ on most datasets. A controller network based on random weights ($DAN_{noise}$) works quite well given that its number of learned parameters is a fifth of the other methods. $DAN_{imagenet}$: controller networks initialized from VGG-B model pretrained on ImageNet. $DAN_{imagenet+sketch}$: selective control network based on both VGG-B(P) & Sketch. We color code the first, second and third highest values in each column (lowest for #par). #par: amortized number of weights learned to achieve said performance for all tasks divided by number of tasks addressed (lower is better).

when applied on the VGG-B architecture. While our method can be applied to any network with convolutional layers, if $C_o \geq D$, i.e., the number of output filters is greater than the dimension of each input filter, it would only increase the number of parameters.

## 4 EXPERIMENTS

We conduct several experiments to test our method and explore different aspects of its behavior. We use two different basic network architectures on two (somewhat overlapping) sets of classification benchmarks. The first is VGG-B (Simonyan and Zisserman (2014)) which we use for various analyses of our method. We begin by listing the datasets we used (4.0.1), followed by establishing baselines by training a separate model for each using a few initial networks. We proceed to test several variants of our proposed method (4.1) as well as testing different training schemes. Next, we discuss methods of predicting how well a network would fare as a base-network (4.2). We show how to discern the domain of an input image and output a proper classification (4.2.1) without manual choice of the control parameters $\alpha$. In the second part of our experiments, we show results on the Visual Decathlon Challenge, using a different architecture. Before concluding we show some more useful properties of our method.

### 4.0.1 DATASETS AND EVALUATION

The first part of our evaluation protocol resembles that of Bilen and Vedaldi (2017): we test our method on the following datasets: **Caltech-256** (Griffin et al. (2007)), **CIFAR-10** (Krizhevsky and Hinton (2009)), **Daimler** (**Munder and Gavrila (2006)**) (DPed), **GTSR** (Stallkamp et al. (2012)), **Omniglot** (**Mnih et al. (2015)**), **Plankton imagery data** (**Cowen et al. (2015)**) (Plnk), **Human Sketch dataset** (Eitz et al. (2012)) and **SVHN** (**Netzer et al. (2011)**). All images are resized to 64 × 64 pixels, duplicating gray-scale images so that they have 3 channels as do the RGB ones. We whiten all images by subtracting the mean pixel value and dividing by the variance per channel. This is done for each dataset separately. We select 80% for training and 20% for validation in datasets where no fixed split is provided. We use the B architecture described in Simonyan and Zisserman (2014), henceforth referred to as VGG-B. It performs quite well on the various datasets when trained from scratch (See Tab. 1). Kindly refer to Bilen and Vedaldi (2017) for a brief description of each dataset.

As a baseline, we train networks independently on each of the 8 datasets. All experiments in this part are done with the Adam optimizer (Kingma and Ba (2014)), with an initial learning rate of 1e-3 or 1e-4, dependent on a few epochs of trial on each dataset. The learning rate is halved after each 10 epochs. Most networks converge within the first 10-20 epochs, with mostly negligible improvements

afterwards. We chose Adam for this part due to its fast initial convergence with respect to non-adaptive optimization methods (e.g, SGD), at the cost of possibly lower final accuracy (Wilson et al. (2017)). The top-1 accuracy (%) is summarized in Tab. 1.

## 4.1 CONTROLLER NETWORKS

To test our method, we trained a network on each of the 8 datasets in turn to be used as a base network for all others. We compare this to the baseline of training on each dataset from scratch (**VGG-B(S)**) or pretrained (**VGG-B(P)**) networks. Tab. 1 summarizes performance on all datasets for two representative base nets: $DAN_{caltech-256}$ (79.9%) and $DAN_{sketch}$ (83.7%). Mean performance for other base nets are shown in Fig. 2 (a). The parameter cost (3.1) of each setting is reported in the last column of the table. This (similarly to Rebuffi et al. (2017)) is the total number of parameters required for a set of tasks normalized by that of a single fully-parametrized network. We also check how well a network can perform as a base-network after it has seen ample training examples: $DAN_{imagenet}$ is based on VGG-B pretrained on ImageNet (Russakovsky et al. (2015)). This improves the average performance by a significant amount (83.7% to 86.5%). On Caltech-256 we see an improvement from 88.2% (training from scratch). However, for both Sketch and Omniglot the performance is in favor of $DAN_{sketch}$. Note these are the only two domains of strictly unnatural images. Additionally, $DAN_{imagenet}$ is still slightly inferior to the non-pretrained **VGG-B(S)** (86.5% vs 87.7%), though the latter is more parameter costly.

**Multiple Base Networks** A good base network should have features generic enough so that a controller network can use them for any target task. In practice this is not necessarily the case. To use two base-networks simultaneously, we implemented a dual-controlled network by using both $DAN_{caltech-256}$ and $DAN_{sketch}$ and attaching to them controller networks. The outputs of the feature parts of the resulting sub-networks were concatenated before the fully-connected layer. This resulted in the exact same performance as $DAN_{sketch}$ alone. However, by using selected controller-modules per group of tasks, we can improve the results: for each dataset the maximally performing network (based on validation) is the basis for the control module, i.e., we used $DAN_{imagenet}$ for all datasets except Omniglot and Sketch. For the latter two we use $DAN_{sketch}$ as a base net. We call this network $DAN_{imagenet+sketch}$. At the cost of more parameters, it boosts the mean performance to **87.76**% - better than using any single base net for controllers or training from scratch. Since it is utilized for 9 tasks (counting ImageNet), its parameter cost (2.76) is still quite good.

**Starting from a Randomly Initialized Base Network** We tested how well our method can perform without any prior knowledge, e.g., building a controller network on a *randomly initialized base network*. The total number of parameters for this architecture is 12M. However, as 10M have been randomly initialized and only the controller modules and fully-connected layers have been learned, the effective number is actually 2M. Hence its parameter cost is determined to be 0.22. We summarize the results in Tab. 1. Notably, the results of this initialization worked surprisingly well; the mean top-1 precision attained by this network was 76.3%, slightly worse than of $DAN_{caltech-256}$ (79.9%). This is better than initializing with $DAN_{daimler}$, which resulted in a mean accuracy of 75%. This is possible the random values in the base network can still be linearly combined by our method to create ones that are useful for classification.

### 4.1.1 INITIALIZATION

One question which arises is how to initialize the weights $W$ of a control-module. We tested several options: (1) Setting $W$ to an identity matrix (**diagonal**). This is equivalent to the controller module starting with a state which effectively mimics the behavior of the base network (2) Setting $W$ to random noise (**random**) (3) Training an independent network for the new task from scratch, then set $W$ to best linearly approximate the new weights with the base weights (**linear_approx**). To find the best initialization scheme, we trained $DAN_{sketch \rightarrow caltech256}$ for one epoch with each and observed the loss. Each experiment was repeated 5 times and the results averaged. From Fig. 2(a), it is evident that the **diagonal** initialization is superior: perhaps counter-intuitively, there is no need to train a fully parametrized target network. Simply starting with the behavior of the base network and tuning it via the control modules results in faster convergence. Hence we train controller modules with the diagonal method. Interestingly, the residual adaptation unit in Rebuffi et al. (2017) is initially similar to the diagonal configuration. If all of the filters in their adapter unit are set to 1 (up

| Dataset | DPed | SVHN | GTSR | C-10 | Oglt | Plnk | Sketch | Caltech |
|---|---|---|---|---|---|---|---|---|
| ft-full | 60.1 | 60 | 65.8 | 70.9 | 80.4 | 81.6 | **84.2** | 82.3 |
| ft-full-bn-off | 61.8 | 64.6 | 66.2 | 72.5 | 78 | 80.2 | **82.5** | 81 |
| ft-last | 24.4 | 33.9 | 42.5 | 44 | 44.1 | 47 | 50.3 | **55.6** |

Table 2: Mean transfer learning performance. We show the mean top-1 accuracy (%) attained by fine-tuning a network from each domain to all domains. Out of the datasets above, starting with Caltech-256 proves most generic as a feature extractor (*ft-last*). However, fine tuning is best when initially training on the Sketch dataset (*ft-full*).

to normalization), the output of the adapter will be initially the same as that of the controller unit initialized with the identity matrix.

## 4.2 TRANSFERABILITY

How is one to choose a good network to serve as a base-network for others? As an indicator of the representative power of the features of each independently trained network $N$, we test the performance on other datasets, using $N$ for fine tuning. We define the *transferability* of a source task $S$ w.r.t a target task $T$ as the top-1 accuracy attained by fine-tuning $N$ trained on $S$ to perform on $T$. We test 3 different scenarios, as follows: (1) Fine-tuning only the last layer (a.k.a feature extraction) (**ft-last**); (2) Fine-tuning all layers of $N$(**ft-full**); (3) same as ft-full, but freezing the parameters of the batch-normalization layers - this has proven beneficial in some cases - we call this option **ft-full-bn-off**. The results in Fig 1 (b) show some interesting phenomena. First, as expected, feature extraction (ft_last) is inferior to fine-tuning the entire network. Second, usually training from scratch is the most beneficial option. Third, we see a distinction between natural images (Caltech-256, CIFAR-10, SVHN, GTSR, Daimler) and unnatural ones (Sketch, Omniglot, Plankton); Plankton images are essentially natural but seem to exhibit different behavior than the rest. It is evident that features from the natural images are less beneficial for the unnatural images. Interestingly, the converse is not true: training a network starting from Sketch or Omniglot works quite well for most datasets, both natural and unnatural. This is further shown in Tab. 2 (a): we calculate the mean transferability of each dataset by the mean value of each rows of the transferability matrix from Fig. 1. $DAN_{Caltech-256}$ works best for feature extraction. However, for full fine-tuning using $DAN_{Plankton}$ works as the best starting point, closely followed by $DAN_{Caltech-256}$. For controller networks, the best mean accuracy attained for a single base net trained from scratch is attained using $DAN_{sketch}$ (83.7%). This is close to the performance attained by full transfer learning from the same network (84.2%, see Tab. 2) at a fraction of the number of parameters. This is consistent with our transferability measure. To further test the correlation between the transferability and the performance given a specific base network, we used each dataset as a base for control networks for all others and measured the mean overall accuracy. The results can be seen in Fig. 2 (b).

### 4.2.1 A UNIFIED NETWORK

Finally, we test the possibility of a single network which can both determine the domain of an image and classify it. We train a classifier to predict from which dataset an image originates, using the training images from the 8 datasets. This is learned easily by the network (also VGG-B) which rapidly converges to 100% or near accuracy. With this "dataset-decider", named $N_{dc}$ we augment $DAN_{sketch}$ to set for each input image $I$ from any of the datasets $D_i$ the controller scalar $\alpha_i$ of $DAN_{sketch \to D_i}$ to 1 if and only if $N_{dc}$ deemed $I$ to originate from $D_i$ and to 0 otherwise. This produces a network which applies to each input image the correct controllers, classifying it within its own domain.

## 4.3 VISUAL DECATHLON CHALLENGE

We now show results on the recent Visual Decathlon Challenge of Rebuffi et al. (2017). The challenge introduces involves 10 different image classification datasets: **ImageNet** (Russakovsky et al. (2015)); **Aircraft** (Maji et al. (2013)); **Cifar-100** (Krizhevsky and Hinton (2009)); **Daimler Pedes-**

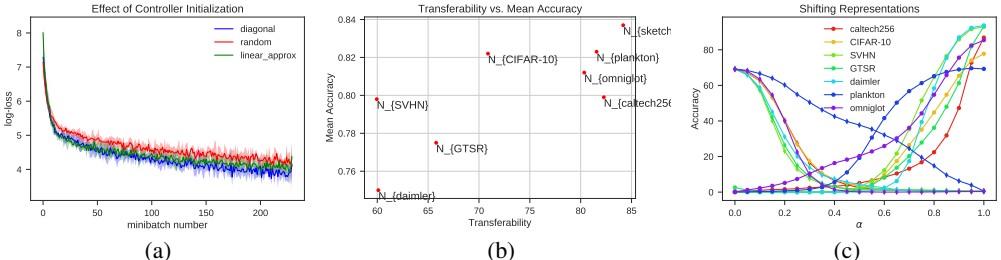

(a)            (b)            (c)

Figure 2: (*a*) Controller initialization schemes. Mean loss averaged over 5 experiments for different ways of initializing controller modules, overlaid with minimal and maximal values. Random initialization performs the worst (random). Approximating the behavior of a fine-tuned network is slightly better (linear_approx) and initializing by mimicking the base network (diagonal) performs the best (*b*) Predictability of a control network's overall accuracy average over all datasets, given its transferability measure. (c) Shifting Representations. Using a single base network $N_{sketch}$, we check the method's sensitivity to varying values of $\alpha$ by varying it in the range $[0, 1]$. Increasing $\alpha$ shifts the network away from the base representation and towards learned tasks - gradually lowering performance on the base task (diamonds) and improving on the learned ones (full circles). The relatively slow decrease of the performance on sketch (blue diamonds) and increase in that of Plankton (blue circles) indicates a similarity between the learned representations.

**trians** (Munder and Gavrila (2006)); **Dynamic Textures** (Cimpoi et al. (2013)); **GTSR** (Stallkamp et al. (2012)); **Flowers** (Nilsback and Zisserman (2008)); **Omniglot** (Lake et al. (2015)); **SVHN** (Netzer et al. (2011)) and **UCF-101** (Soomro et al. (2012)). The goal is to reach accurate classification on each dataset while retaining a small model size, using the train/val/test splits fixed by the authors. All images are resized so the smaller side of each image is 72 pixels. The classifier is expected to operate on images of 64x64 pixels. Each entry in the challenge is assigned a decathlon score, which is a function designed to highlight methods which do better than the baseline on all 10 datasets. Please refer to the challenge website for details about the scoring and datasets: `http://www.robots.ox.ac.uk/~vgg/decathlon/`. Similarly to Rebuffi et al. (2017), we chose to use a wide residual network (Zagoruyko and Komodakis (2016)) with an overall depth of 28 and a widening factor of 4, with a stride of 2 in the convolution at the beginning of each basic block. In what follows we describe the challenge results, followed by some additional experiments showing the added value of our method in various settings. In this section we used the recent YellowFin optimizer (Zhang et al. (2017)) as it required less tuning than SGD. We use an initial learning rate factor of 0.1 and reduce it to 0.01 after 25 epochs. This is for all datasets with the exception of ImageNet which we train for 150 epochs with SGD with an initial learning rate of 0.1 which is reduced every 35 epochs by a factor of 10. This is the configuration we determined using the available validation data which was then used to train on the validation set as well (as did the authors of the challenge) and obtain results from the evaluation server. Here we trained on the reduced resolution ImageNet from scratch and used the resulting net as a base for all other tasks. Tab. 3 summarizes our results as well as baseline methods and the those of Rebuffi et al. (2017), all using a base architecture of similar capacity. By using a significantly stronger base architecture they obtained higher results (mean of 79.43%) but with a parameter cost of 12. All of the rows are copied from Rebuffi et al. (2017), including their re-implementation of LWF, except the last which shows our results. The final column of the table shows the decathlon score. A score of 2500 reflects the baseline: finetuning from ImageNet for each dataset independently. For the same architecture, the best results obtained by the Residual Adapters method is slightly below ours in terms of decathlon score and slightly above them in terms of mean performance. However, unlike them, we avoid joint training over all of the datasets and using dataset-dependent weight decay.

### 4.4 COMPRESSION AND CONVERGENCE

In this section, we highlight some additional useful properties of our method. All experiments in the following were done using the same architecture as in the last section but training was performed only on the training sets of the Visual Decathlon Challenge and tested on the validation sets. First,

| Method | #par | ImNet | Airc. | C100 | DPed | DTD | GTSR | Flwr | Oglt | SVHN | UCF | mean | S |
|---|---|---|---|---|---|---|---|---|---|---|---|---|---|
| Scratch | 10 | 59.87 | 57.1 | 75.73 | 91.2 | 37.77 | 96.55 | 56.3 | 88.74 | 96.63 | 43.27 | 70.32 | 1625 |
| Feature | 1 | 59.67 | 23.31 | 63.11 | 80.33 | 45.37 | 68.16 | 73.69 | 58.79 | 43.54 | 26.8 | 54.28 | 544 |
| Finetune | 10 | 59.87 | 60.34 | 82.12 | 92.82 | 55.53 | 97.53 | 81.41 | 87.69 | 96.55 | 51.2 | 76.51 | 2500 |
| LWF | 10 | 59.87 | 61.15 | 82.23 | 92.34 | 58.83 | 97.57 | 83.05 | 88.08 | 96.1 | 50.04 | 76.93 | 2515 |
| Res. Adapt. | 2 | 59.67 | 56.68 | 81.2 | 93.88 | 50.85 | 97.05 | 66.24 | 89.62 | 96.13 | 47.45 | 73.88 | 2118 |
| Res. Adapt (Joint) | 2 | 59.23 | 63.73 | 81.31 | 93.3 | 57.02 | 97.47 | 83.43 | 89.82 | 96.17 | 50.28 | **77.17** | 2643 |
| DAN (Ours) | 2.17 | 57.74 | 64.12 | 80.07 | 91.3 | 56.54 | 98.46 | 86.05 | 89.67 | 96.77 | 49.38 | 77.01 | **2851** |

Table 3: Results on Visual Decathlon Challenge. *Scratch*: training on each task independently. *Feature*: using a pre-trained network as a feature extractor. *Finetune* : vanilla fine tuning. Performs well but requires many parameters. Learning-without-forgetting (*LWF*, Li and Hoiem (2016)) slightly outperforms it but with a large parameter cost. *Residual adapt.* (Rebuffi et al. (2017)) significantly reduce the number of parameters. Results improve when training jointly on all task (*Res.Adapt(Joint)*). The proposed method (DAN) outperforms residual adapters despite adding each task *independently* of the others. **S** is the decathlon challenge score.

we check whether the effects of network compression are complementary to ours or can hinder them. Despite the recent trend of sophisticated network compression techniques (for example Han et al. (2015a)) we use only a simple method of compression as a proof-of-concept, noting that using recent compression methods will likely produce better results. We apply a simple linear quantization on the network weights, using either 4, 6, 8, 16 or 32 bits to represent each weight, where 32 means no quantization. We do not quantize batch-normalization coefficients. Fig. 3 (b) shows how accuracy is affected by quantizing the coefficients of each network. Using 8 bits results in only a marginal loss of accuracy. This effectively means our method can be used to learn new tasks with a cost of 3.25% of the original parameters. Many maintain performance even at 6 bits (DPed, Flowers, GTSR, Omniglot, SVHN. Next, we compare the effect of quantization on different transfer methods: feature extraction, fine-tuning and our method (both diagonal and linear variants). For each dataset we record the normalized (divided by the max.) accuracy for each method/quantization level (which is transformed into the percentage of required parameters). This is plotted in Fig. 4 (a). Our method requires significantly less parameters to reach the same accuracy as fine-tuning. If parameter usage is limited, the diagonal variant of our method significantly outperforms feature extraction. Finally, we show that the number of epochs until nearing the maximal performance is markedly lower for our method. This can be seen in Fig. 4 (b,c).

## 4.5 Discussion

We have observed that the proposed method converges to a a reasonably good solution faster than vanilla fine-tuning and eventually attains slightly better performance. This is despite the network's expressive power, which is limited by our construction. We conjecture that constraining each layer to be expressed as a linear combination of the corresponding layer in the original network serves to regularize the space of solutions and is beneficial when the tasks are sufficiently related to each other. One could come up with simple examples where the proposed method would likely fail: if the required solutions to two tasks are disjoint. For example, one task requires counting of horizontal lines and the other requires counting of vertical ones, and such examples are all that appear in the training sets, then the proposed method will likely work far worse than vanilla fine-tuning or training from scratch. We leave the investigation of this issue, as well as finding ways between striking a balance between reusing features and learning new ones as future work.

## 5 Conclusions

We have presented a method for transfer learning thats adapts an existing network to new tasks while fully preserving the existing representation. Our method matches or outperforms vanilla fine-tuning, though requiring a fraction of the parameters, which when combined with net compression

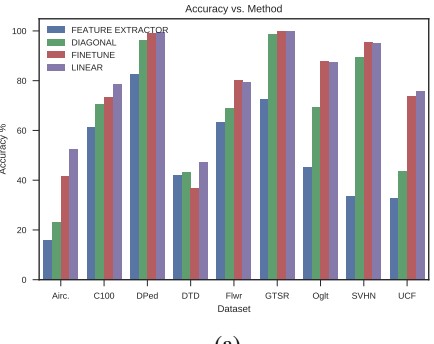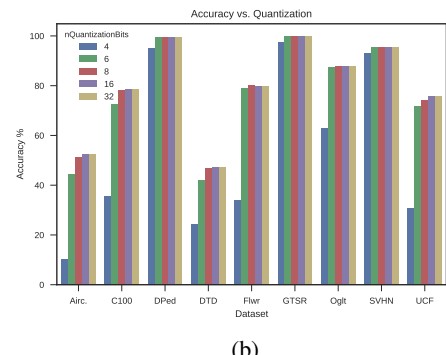

Figure 3: (a) Accuracy vs. learning method. Using only the last layer (*feature extractor*) performs worst. *finetune*: vanilla fine-tuning. *Diagonal* : our controller modules with a diagonal combination matrix. *Linear*: our full method. On average, our full method outperforms vanilla fine tuning. (b) Accuracy vs. quantization: with as low as 8 bits, we see no significant effect of network quantization on our method, showing they can be applied together.

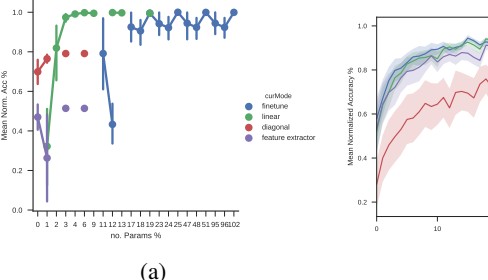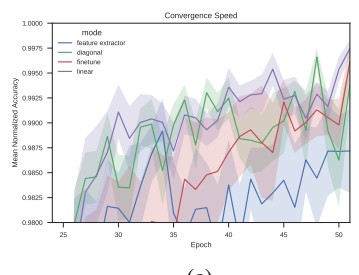

Figure 4: (a) Mean classification accuracy (normalized, averaged over datasets) w.r.t no. parameters. Our method achieve better performance over baselines for a large range of parameter budgets. For very few parameters diagonal (ours) outperforms features extraction. To obtain maximal accuracy our full method requires far fewer parameters (see linear vs finetune). (b) Our method (*linear*) converges to a high accuracy faster than fine-tuning. The weaker variant of our method converges as fast as feature-extraction but reaches an overall higher accuracy (3 (a)). (c) zoom in on top-right of (b).

reaches 3% of the original parameters with no loss of accuracy. The method converges quickly to high accuracy while being on par or outperforming other methods with the same goal. Built into our method is the ability to easily switch the representation between the various learned tasks, enabling a single network to perform seamlessly on various domains. The control parameter $\alpha$ can be cast as a real-valued vector, allowing a smooth transition between representations of different tasks. An example of the effect of such a smooth transition can be seen in Fig. 2 (c) where $\alpha$ is used to linearly interpolate between the representation of differently learned tasks, allowing one to smoothly control transitions between different behaviors. Allowing each added task to use a convex combination of already existing controllers will potentially utilize controllers more efficiently and decouple the number of controllers from the number of tasks.

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
