# OpenReview forum: "Incremental Learning through Deep Adaptation"
_ICLR.cc/2018/Conference — Reject_

### Official Review · AnonReviewer2 · 2017-11-27
**-**

**Rating:** 6
**Confidence:** 4

**Review:**

This paper proposes to adapt convnet representations to new tasks while avoiding catastrophic forgetting by learning a per-task “controller” specifying weightings of the convolution-al filters throughout the network while keeping the filters themselves fixed.


Pros

The proposed approach is novel and broadly applicable.  By definition it maintains the exact performance on the original task, and enables the network to transfer to new tasks using a controller with a small number of parameters (asymptotically smaller than that of the base network).

The method is tested on a number of datasets (each used as source and target) and shows good transfer learning performance on each one.  A number of different fine-tuning regimes are explored.

The paper is mostly clear and well-written (though with a few typos that should be fixed).


Cons/Questions/Suggestions

The distinction between the convolutional and fully-connected layers (called “classifiers”) in the approach description (sec 3) is somewhat arbitrary -- after all, convolutional layers are a generalization of fully-connected layers. (This is hinted at by the mention of fully convolutional networks.)  The method could just as easily be applied to learn a task-specific rotation of the fully-connected layer weights.  A more systematic set of experiments could compare learning the proposed weightings on the first K layers of the network (for K={0, 1, …, N}) and learning independent weights for the latter N-K layers, but I understand this would be a rather large experimental burden.

When discussing the controller initialization (sec 4.3), it’s stated that the diagonal init works the best, and that this means one only needs to learn the diagonals to get the best results.  Is this implying that the gradients wrt off-diagonal entries of the controller weight matrix are 0 under the diagonal initialization, hence the off-diagonal entries remain zero after learning?  It’s not immediately clear to me whether this is the case -- it could help to clarify this in the text.

If the off-diag gradients are indeed 0 under the diag init, it could also make sense to experiment with an “identity+noise” initialization of the controller matrix, which might give the best of both worlds in terms of flexibility and inductive bias to maintain the original representation. (Equivalently, one could treat the controller-weighted filters as a “residual” term on the original filters F with the controller weights W initialized to noise, with the final filters being F+(W\crossF) rather than just W\crossF.)

The dataset classifier (sec 4.3.4) could be learnt end-to-end by using a softmax output of the dataset classifier as the alpha weighting. It would be interesting to see how this compares with the hard thresholding method used here.  (As an intermediate step, the performance could also be measured with the dataset classifier trained in the same way but used as a soft weighting, rather than the hard version rounding alpha to 0 or 1.)


Overall, the paper is clear and the proposed method is sensible, novel, and evaluated reasonably thoroughly.

---

> ### Author Response · Authors · 2017-12-04
> **Please see above answer**
>
> It contains replies to all reviewer's comments.

---

### Official Review · AnonReviewer3 · 2017-11-28
**Interesting idea but missing some simple baselines.**

**Rating:** 4
**Confidence:** 4

**Review:**

This paper proposes new idea of using controller modules for increment learning. Instead of finetuning the whole network, only the added parameters of the controller modules are learned while the output of the old task stays the same. Experiments are conducted on multiple image classification datasets.

I found the idea of using controller modules for increment learning interesting and have some practical use cases. However, this paper has the following weakness:
1) Missing simple baselines. I m curious to see some other multitask learning approach, e.g. branch out on the last few layers for different tasks and finetune the last few layers. The number of parameters won't be affected so much and it will achieve better performance than 'feature' in table 3.
2) Gain of margin is really small. The performance improvements in Table1 and Table3 are very small. I understand the point is to argue with fewer parameters the model can achieve comparable accuracies. However, there could be other ways to design the network architecture to reduce the size (sharing the lower level representations).
3) Presentation of the paper is not quite good. Figures are blurry and too small.

---

> ### Author Response · Authors · 2017-12-04
> **Please see above answer.**
>
> It contains replies to all reviewer's comments.

---

### Official Review · AnonReviewer1 · 2017-11-29
**Extensive experiments but inconclusive for the main message (task-incremental learning)**

**Rating:** 5
**Confidence:** 4

**Review:**

----------------- Summary -----------------
The paper tackles the problem of task-incremental learning using deep networks. It devises an architecture and a training procedure aiming for some desirable properties; a) it does not require retraining using previous tasks’ data, b) the number of network parameters grows only sublinearly c) it preserves the output of the previous tasks intact.

----------------- Overall -----------------
The paper tackles an important problem, aims for important characteristics, and does extensive and various experiments. While the broadness of the experiments are encouraging, the main task which is to propose an effective task-incremental learning procedure is not conclusively tested, mainly due to the lack of thorough ablation studies (for instance when convolutional layers are fixed) and the architecture seems to change from one baseline (method) to another.

----------------- Details -----------------
- in the abstract it says: "Existing approaches either learn sub-optimal solutions, require joint training, or incur a substantial increment in the number of parameters for each added task, typically as many as the original network."
The linear-combination constraint in the proposed approach is a strong one and can learn a sub-optimal solution for the newly introduced tasks.

- Page 3: R^C → R^{C_o}

- The notation is (probably unnecessarily) too complicated, perhaps it’s better to formulate it without being faithful to the actual implementation but for higher clarity and ease of understanding. For instance, one could start from denoting feature maps and applying the controller/transform matrix W on that, circumventing the clutter of convolutional kernels.

- What is the DAN architecture?

- In table 1 a better comparison is when using same architecture (instead of VGG) to train it from scratch or fine-tune from ImageNet (the first two rows)

- What is the architecture used for random-weights baseline?

- An experiment is needed where no controller is attached but just the additional fully-connected layers to see the isolated improvements gained by the linear transform of convolutional layers.

- Multiple Base Networks: The assumption in incremental learning is that one does not have access to all tasks/datasets at once, otherwise one would train them jointly which would save parameters, training time and performance. So, finding the best base network using the validation set is not relevant.

- The same concern as above applies to the transferability and dataset decider experiments

---

### Author Response · Authors · 2017-12-04
**Replies to reviewer's comments**

Thanks for the time taken for reviewing an the constructive suggestions.
Replies to reviewers:
Reviewer 1:

1. Perhaps some additional ablation studies as freezing some layers are in place, though several baselines were tested, ranging from freezing all but the top-layer ("feature"), freezing nothing ("fine-tuning"), as well as comparing (see table 3) to stronger baselines such as LWF and the very recent Residual Adapters - all of which are outperformed. Some experiments were omitted due to lack of space and to avoid missing the main point due to cluttering the paper. In addition, less powerful variants of the proposed method were suggested and evaluated, such as the "diagonal" method.
2. Indeed two main architectures were used, the VGG architecture for exploratory experiments regarding transferability, initialization methods, etc and for the Visual Decathlon Challenge a Res-net based architecture was used. Perhaps the exposition or order of experiments caused confusion. However, using different datasets and architectures also serves to show applicability of the across settings.
3.  About sub-optimality: indeed we constrain the space of solutions. A task whose basic required features are span an orthogonal subspace will surely result in poor performance under this method. This limitation is quite explicitly acknowledged in the discussion (section 4.5) and mentioned as an issue for future work. In addition,  we address these issues in the experiments by testing which base-network is suitable for transferring to other tasks with the best average performance (see Fig 1(b), as well as section 4.2, and fig. 2(b)). Arguably, also the number of convolutional filters in the first level of a modern CNN is limiting, for example, in resnet we have 16 3x3 filters, and the space of 3x3 RGB channels would require 27 3x3 filters to be fully spanned. But we know that the space of natural images is much smaller than that of all images (though likely not a linear subspace).
4. Notation : We weren't sure if explicitly writing the notation this was would be better or worse than leaving it in a more compact form. We agree it seems a bit over-complicated.
5. DAN refers to any architecture which was augmented with the controller modules + extra heads for additional tasks. We regret this was not clear from the text and can try to clarify it.
6. What better comparison would you suggest for table 1? This captures both transferability or powerful pre-training w.r.t various tasks and the compactness of representation.
6. "an experiment is needed with just an additional fully connected layer" : this is actually in the paper as one of the baselines, e.g. called "feature" in table 3, also referred to as "feature extraction", "shallow transfer learning" , "ft-last" (table 2, 3rd row).
7. Multiple base-networks: We agree with the reviewer's reasoning if all the data is not available at once. This, as well as the transferability tests, were more of an exploratory nature to see relations between representations learned on various dataset.

Reviewer3:
1. Please see answer 1 to reviewer 1.
2. We do not claim that performance in terms of accuracy is much higher than regular
fine-tuning. The main claim is indeed efficiency of representation and this is not left
without comparison to several other methods, including recent ones; As shown in table 3,
we outperform LWF - though not by a large margin, but do gain much in terms of representation size, and outperform the incremental version of Residual adapters, and match Residual adapters where they used *joint* training. To recap, we show improvements over well accepted and some very recent baselines that address some of the same challenges.
3. Please elaborate on what you mean by "presentation". Does this refer to figure aesthetics? Indeed, some can be made larger. Which figures were blurry?

Reviewer2:
1. The reason to avoid tasks-specific rotation of fc layers is because the number of weights required to do so would usually surpass that required to learn all parameters anew, e.g, a fc layer of 512x1000 would require 1000x1000 parameters.
2. About diagonal init: this is discussed (though not in terms of gradients) in section 4.1.1, and indeed a the text briefly mentions a similar recent work that does as the reviewer suggested by using residual units.
3. We had some very initial experiments with soft thresholding / weighing but these were left out, as the paper was already quite long. There is mention on shifting representations in a soft way, see Fig. 2(c). The suggestion of training with a non-integer alpha is a very interesting one and is reminiscent of recent work on training with affine-combinations training images. Thanks for the suggestion!

---

### Decision · Program_Chairs · 2018-01-29
**ICLR 2018 Conference Acceptance Decision**

**Decision:**

Reject

**Comment:**

This work tackles an important problem of incremental learning and does so with extensive experimentation. As pointed out by two reviewers, the idea does seem novel and interesting, but the submission would require some rewriting before being potentially accepted at a venue like ICLR. I suggest focusing the paper more on the task-incremental learning aspects, doing the ablation studies (and other changes) as requested by the reviewers, and having a rich appendix with details (with more discussion in the paper itself).